# Psychometric Properties of a Short Academic Motivation Scale (SAMS) in Medical Students

**DOI:** 10.3390/bs14040316

**Published:** 2024-04-12

**Authors:** Jai Pascual-Mariño, Mardel Morales-García, Liset Z. Sairitupa-Sanchez, Oscar Mamani-Benito, Percy G. Ruiz Mamani, Sandra B. Morales-García, Oriana Rivera-Lozada, Wilter C. Morales-García

**Affiliations:** 1Unidad de Ciencias Humanas y Educación, Escuela de Posgrado, Universidad Peruana Unión, Lima 15457, Peru; 2Unidad de Ciencias de la Salud, Escuela de Posgrado, Universidad Peruana Unión, Lima 15457, Peru; 3Escuela Profesional de Psicología, Facultad de Ciencias de la Salud, Universidad Peruana Unión, Lima 15457, Peru; 4Facultad de Ciencias de la Salud, Universidad Señor de Sipán, Chiclayo 14001, Peru; 5Escuela de Enfermería, Universidad Privada San Juan Bautista, Lima 15457, Peru; 6Departamento Académico de Enfermería, Obstetricia y Farmacia, Facultad de Farmacia y Bioquímica, Universidad Científica del Sur, Lima 15457, Peru; 7Facultad de Educación, Universidad Nacional Mayor de San Marcos, Lima 15457, Peru; 8Escuela de Medicina Humana, Facultad de Ciencias de la Salud, Universidad Peruana Unión, Lima 15457, Peru; 9Facultad de Teología, Universidad Peruana Unión, Lima 15457, Peru; 10Sociedad Científica de Investigadores Adventistas, SOCIA, Universidad Peruana Unión, Lima 15457, Peru

**Keywords:** academic, motivation, medical students, SAMS, psychometric properties

## Abstract

***Background***: Medical education represents a complex field of study, influenced by various psychological, demographic, and contextual factors. Academic motivation, essential for educational success, has been linked to critical decisions in medical careers and can be modulated by contextual elements such as socioeconomic and geographical environments. The theory of self-determination has provided a solid framework for understanding the multidimensional nature of motivation. ***Objective***: To evaluate the psychometric properties of a Spanish version of the Short Scale of Academic Motivation among Peruvian medical students. ***Methods***: Using an instrumental design, the factorial structure, reliability, and gender invariance of the SAMS-S were assessed. A confirmatory factor analysis (CFA) was conducted to validate the scale’s structure based on seven dimensions. Additionally, reliability was assessed through Cronbach’s alpha coefficient and omega coefficient, and gender invariance was determined through multi-group confirmatory factor analysis. ***Results***: The Peruvian version of the SAMS-S showed a good fit in the CFA with satisfactory goodness-of-fit indices. However, challenges in discriminant validity among certain dimensions were detected, suggesting the presence of a second-order factor. The proposed second-order model yielded an adequate fit (χ^2^ = 198.26, df = 70, CFI = 0.92, TLI = 0.90, RMSEA = 0.08 [90% CI: 0.07–0.1], SRMR = 0.07), validating the factorial structure of the SAMS-S. The scale’s reliability and its subscales were within acceptable ranges. Furthermore, the gender invariance of the SAMS-S was confirmed at all levels, from configural to strict. ***Conclusions***: The second-order model of the SAMS-S presents as a valid and reliable tool for measuring academic motivation among medical students in Peru. Its robustness and adaptability make it relevant for future research in similar educational contexts and can serve as a basis for interventions aimed at improving academic motivation in this specific group.

## 1. Introduction

Medical education is a vital field for the advancement of global health, characterized by its dynamism and complexity. It is influenced by factors as diverse as technological advances and political challenges in healthcare, underscoring the importance of aligning academic competencies with the practical demands of patient care [1,2]. The ongoing evolution of this field demands highly motivated and committed students, whose education is analyzed through theoretical approaches ranging from cognitivism to constructivism, highlighting the relevance of both intrinsic and extrinsic motivation in their academic performance and professional socialization [3,4,5].

Academic motivation, far from being an isolated phenomenon, interacts with psychological and environmental aspects such as self-efficacy, professional expectations, and the educational environment. It is also influenced by demographic factors like gender, academic year, and geographical location [5,6,7,8]. In the medical field, motivation is crucial in key decisions such as specialty choice, showing significant variations according to regional and socioeconomic contexts [9,10,11]. The changing perception of medicine, moving away from its traditional status of prestige, highlights the need for a deeper understanding of the motivational elements that drive students towards robust training and an effective transition to professional life [12,13].

In medical sciences, academic motivation is fundamental as it not only defines a student’s persistence in their university career but also their active participation in learning [14,15,16]. Achievement motivation and academic motivation, while distinct in nature, underscore the importance of self-determination and adaptability in the educational process [17,18]. The theory of self-determination (SDT) provides a comprehensive framework for understanding motivation [19] as a spectrum that ranges from intrinsic motivation, through extrinsic motivation, to amotivation, with intrinsic motivation highlighted as the purest form of motivation, derived from the pleasure and satisfaction inherent in learning [20,21,22]. This deep understanding of academic motivation underscores its critical role in student engagement, performance, and retention in education [18].

Among Peruvian students, academic motivation reveals complexities inherent in the educational process across various areas of study. In this sense, both intrinsic motivation, linked to a personal desire to learn, and extrinsic motivation, associated with external rewards such as job opportunities, influence academic performance, albeit with weak correlations [23]. In contexts such as that of international business students, social pressure and personal aspirations play a significant role in career choice, differing from fields like health, where motivation for public service predominates [24]. In the field of medicine, the trend indicates high motivation, particularly notable among women, despite the challenges presented by the transition to virtual education during the COVID-19 pandemic. This underscores the need for further research to thoroughly understand the factors affecting motivation in medical education scenarios [25]. Additionally, adaptation to university life emerges as a significant challenge, where elements such as procrastination, self-esteem, and self-efficacy are identified as critical determinants for academic success, highlighting the importance of addressing these factors to facilitate better adaptation and student performance [26].

This intricate web of a student’s desire and commitment to academic subjects is assessed when the student’s competence is contrasted with a standard of performance or excellence, and various scales have been developed for this purpose. Among the most notable are the Children’s Academic Intrinsic Motivation Inventory (CAIMI) [27], the Motivated Strategies for Learning Questionnaire (MSLQ) [28], and the Achievement Emotions Questionnaire (AEQ) [29]. However, the Academic Motivation Scale (AMS) [30] stands out due to its psychometric robustness and its focus on intrinsic and extrinsic motivation, grounded in Self-Determination Theory (SDT). The AMS, originally created for Canadian university students, has been adapted and validated in various contexts and populations [31,32,33,34], with a 28-item scale and a confirmed seven-factor factorial structure [30]. Given the importance of having precise tools adapted to the context, and considering the significant performance of the AMS, an initiative for an abbreviated version emerged: the Short Academic Motivation Scale (SAMS) [35]. This scale was originally designed for business and medical students in the United Kingdom. The SAMS is conceived as a response to the need for reducing the response burden, potentially increasing the completion rate and improving the data quality.

Nevertheless, the reality of medical students in Peru could be different, which underscores the necessity of having specific and adapted measures. The SAMS, with its brevity and specificity, could offer an ideal solution for assessing academic motivation among Peruvian medical students. Indeed, a study in Lambayeque revealed that 66% of the students from a private university reported a high level of academic motivation [25]. Given the limited literature on the psychometric assessment of academic motivation in the Peruvian context and the growing importance of medical education and the unique demands these students face, it is essential to have instruments that accurately reflect their motivation levels, allowing for more informed and effective educational interventions. Therefore, the objective of the present research is to evaluate the psychometric properties of the Short Academic Motivation Scale (SAMS) in Peruvian medical students.

## 2. Method

### 2.1. Design and Participants

This is a quantitative cross-sectional and instrumental study [36] (Ato et al., 2013). The inclusion criteria for participants were as follows: (1) being a medical student enrolled in the School of Medicine, (2) being of legal age, and (3) providing informed consent. Additionally, a sample size calculation was performed considering an expected effect size of λ = 0.10, a statistical significance level of α = 0.05, and a statistical power of 1 − β = 0.90, determining that the minimum adequate sample size was 199 participants (Soper, 2023) [37]. During the data preparation and preliminary analysis process, criteria were implemented to ensure the integrity and quality of the information, including the review of missing cases, identification, and exclusion of outliers. In total, 268 students participated, with ages ranging between 18 and 40 years (M = 19.68, SD = 3.01). Of the participants, 59.3% were women and 40.7% men. The majority came from the coastal region (53.7%) and were in their first year (53.0%) (see Table 1).

### 2.2. Instrument

Short Academic Motivation Scale (SAMS). The English version [35] of the Short Academic Motivation Scale (SAMS) assesses a student’s desire (reflected in their focus, persistence, and level of interest) regarding academic subjects when the student’s competence is judged against a standard of performance or excellence. It consists of 14 items and has 7 dimensions: Intrinsic Motivation to Know (IMK), Intrinsic Motivation toward Accomplishment (IMA), Intrinsic Motivation to Experience Stimulation (IMS), Identified Regulation (IDR), Introjected Regulation (IJR), External Regulation (ER), and Amotivation (AM), rated on a seven-point Likert scale (1 = “Does not correspond at all” to 7 = “Corresponds exactly”). The internal consistencies of the SAMS sub-scales range from 0.63 to 0.85, demonstrating acceptable to high reliability.

For the translation of the SAMS instrument into Peruvian Spanish, recommended procedures for the cultural adaptation of instruments were employed [38]. The process began with the initial translation of the SAMS into Spanish by two bilingual Spanish-speaking natives. This translation was then back-translated into English by two English-speaking natives who were not familiar with the content and purpose of the SAMS. The back-translated version was evaluated by a panel comprising three psychologists and two educators, resulting in the development of the SAMS-S version for Peruvian Spanish. To validate the readability and comprehension of this instrument, it was administered to a group of 17 medical students, who faced no comprehension issues (See Table 2).

### 2.3. Procedure

The research took place from 14 February to 12 May 2023, across three Peruvian universities. Prior to data collection, necessary approval from the administrators of these institutions was obtained, ensuring adherence to relevant institutional and ethical policies. The main means of data collection was an online form, meticulously designed to align with the study’s objectives. Once institutional permission was granted, these forms were channeled through the university coordinators, who in turn shared them with the students through institutional platforms. Before accessing the content of the form, students were provided with a detailed explanation of the purpose of the research, the potential benefits of participating, as well as clear assurances about anonymity and confidentiality. The importance of informed consent was highlighted, asking interested students to sign this document beforehand. This consent highlighted the voluntary nature of the research, the minimal associated risks, and the complete protection of the identity of each participant.

### 2.4. Ethics

The research project was evaluated and subsequently approved by the Ethics Committee of a Peruvian university, identified with the code 2023-CEUPeU-019. This approval confirms that the study’s design, methodologies, and ethical considerations are in accordance with international ethical principles, including the Helsinki protocol [39].

### 2.5. Data Analysis

In this study, calculations of descriptive statistics were performed, including mean (M), standard deviation (SD), skewness (g_1_), and kurtosis (g_2_), referencing values within the ±2 range for considerations of normality [40,41]. The data analysis process began by verifying the normality of the distribution using Mardia’s multivariate normality estimation.

A confirmatory factor analysis (CFA) was conducted using the Short Scale of Academic Motivation (SAMS), which encompasses seven dimensions [35]. The robust maximum likelihood estimation (MLR) was chosen due to its robustness in situations of non-normality in the data and the presence of ceiling and floor effects [42,43]. Model fit was assessed using indices such as RMSEA, SRMR, CFI, and TLI. Values below 0.08 for RMSEA and SRMR were considered acceptable, while those under 0.05 indicated an optimal fit [44,45]. For CFI and TLI, values above 0.90 were deemed as adequate and those above 0.95 as indicative of a good fit [46].

The scale’s reliability was measured using Cronbach’s alpha coefficient and the omega coefficient, considering values above 0.70 as adequate [47,48]. Item retention on the scale was based on factor loading values, accepting items with loadings above 0.50 [49]. Convergent validity was estimated by the average variance extracted (AVE), where values above 0.50 are deemed adequate, while discriminant validity was assumed to the extent that the AVE of each latent variable was greater than the square of the correlation (φ^2^) between them [50].

To examine measurement invariance (MI) across gender, a multi-group confirmatory factor analysis was used. Four levels of invariance were assessed: configural, metric, scalar, and strict. ΔCFI differences less than 0.010 were crucial for determining invariance between groups [51]. Additionally, concerning validity with other variables, a model was proposed using structural equation modeling, utilizing the MLR estimator.

For statistical analysis, the RStudio environment (version 4.1.1) was utilized, and packages such as “lavaan” for CFA and structural equation modeling, as well as “semTools” for the analysis of measurement invariance, were employed [52,53].

## 3. Results

### 3.1. Preliminary Analysis

In Table 2, a variety of mean scores and standard deviations for different items related to the SAMS-S are presented. The item that obtained the highest mean (M = 5.54) was “for the pleasure I feel in surpassing one of my personal achievements”, while the item “because I do not know; I cannot understand what I am doing at the university” obtained the lowest mean (M = 2.08). In terms of variability, “because I do not understand why I am going to university and, frankly, I do not care” showed the greatest dispersion in responses (SD = 1.69), whereas the item “because the university allows me to experience personal satisfaction in my pursuit of excellence in my studies” had the least variability (SD = 1.16). Regarding skewness (g_1_), all items, except “because I do not understand why I am going to university and, frankly, I do not care” (g_1_ = 1.17) and “because I do not know; I cannot understand what I am doing at the university” (g_1_ = 1.25), are within the range considered normal (±1.5), suggesting an approximately symmetrical distribution for most. However, both aforementioned items exhibit significant positive skewness. As for kurtosis (g_2_), most items also fall within the range of normality, indicating a typical distribution shape for most, though some items exhibit slight deviations. The results of the multivariate normality analysis, following Mardia’s criteria, clearly indicate that the data do not fit a multivariate normal distribution. This was determined through Mardia’s skewness, which presented a statistical value of 1312.85 and a *p*-value < 0.001, and Mardia’s kurtosis, with a value of 18.09 and a *p*-value of 0. Given this evidence of non-normality, the use of the MLR estimator is recommended, recognized for its robustness in analyzing data that do not follow a normal distribution.

### 3.2. Confirmatory Factor Analysis

Based on the proposed model for the Short Scale of Academic Motivation (SAMS) with seven dimensions [35], the SAMS-S was analyzed. The analysis indicated adequate goodness-of-fit indices: χ^2^ = 132.040, df = 56, *p* = 0.000, CFI = 0.95, TLI = 0.92, RMSEA = 0.07 (90% CI 0.06–0.09), SRMR = 0.04. When examining the factor loadings, all variables exceeded the established criterion of λ > 0.50, indicating that all variables are relevant and have a strong contribution to their respective factors. For convergent validity, all AVEs reached an acceptable magnitude (>0.50), indicating that the constructs capture a robust amount of variance from their corresponding items. However, in terms of internal discriminant validity, the square of the correlation between IMA and IMS exceeded the AVE of IMA. Additionally, IMA shows a similarly high correlation with IJR. The relationship between IMS and IDR also exceeds the threshold, greater than the AVE of IDR. A particularly high correlation is observed between IMS and IJR, exceeding both AVEs, as well as between IDR and IJR. Finally, the correlation between IJR and ER surpasses the AVE of IJR (Table 3). These findings suggest a significant overlap in what these constructs are measuring [54,55], indicating the consideration of a second-order factor influencing these constructs [56,57].

A second model (Figure 1) was evaluated with the aim of gathering evidence to interpret the instrument as a multi-level scale, where a higher-order factor groups the seven dimensions. The performance of the confirmatory factor analysis (CFA) yielded adequate fit indices: χ^2^ = 198.26, df = 70, *p* = 0.000, CFI = 0.92, TLI = 0.90, RMSEA = 0.08 (90% CI: 0.07–0.1), SRMR = 0.07.

### 3.3. Internal Consistency

Internal consistency of the first model was assessed using Cronbach’s alpha coefficient (α), McDonald’s omega (ω), and Composite Reliability (CR) for various subscales. The IMK subscale showed good reliability with values of 0.76 for α, ω, and CR. IMA presented similar values, with 0.71 for α, ω, and CR, indicating consistent reliability. The IMS subscale had values of 0.71 for α and 0.74 for ω and CR, reflecting solid reliability. IDR showed reliability with α of 0.72 and ω and CR of 0.72. IJR had slightly lower values, though close to the threshold of good reliability, with 0.67 for α and 0.69 for ω and CR. The ER subscale exhibited acceptable reliability with α of 0.73 and ω and CR of 0.75. Notably, AM demonstrated excellent reliability with 0.91 for α and 0.92 for ω and CR (Table 3).

Upon evaluating the internal consistency of the second model, the individual subscales display reliability values that reflect good internal consistency, with the IMK subscale showing values of 0.76 for α (Cronbach’s alpha), ω (omega), and CR (Composite Reliability). Similarly, the IMA records consistent values of 0.71 across all three reliability indicators. The IMS subscale stands out with a slight improvement in ω and CR to 0.76. The IDR and IJR show α values of 0.72 and 0.67, respectively, with ω and CR reflecting this trend. The ER exhibits reliability with an α of 0.73 and both ω and CR at 0.74. Notably, AM is particularly robust with α and ω values of 0.91, and a CR of 0.91. Likewise, the overall internal consistency of the model, a critical aspect not previously discussed, yields an α of 0.82, an ω of 0.91, and a CR of 0.86, indicating excellent reliability of the second-order model as a whole (Table 4).

### 3.4. Gender Invariance

The Short Academic Motivation Scale in Spanish (SAMS-S) for medical students underwent a validation process through a series of hierarchical variance models to evaluate its invariance by gender. The analysis began with configural invariance, establishing a baseline model, followed by assessments of metric, scalar, and strict invariance, to determine if the scale’s structure remains consistent across genders. The fit indices, including CFI, TLI, RMSEA, and SRMR, showed good fit across the different levels of invariance. Significantly, the differences in CFI (ΔCFI) were less than 0.01 at all levels [51], indicating that the scale is invariant across genders. That is, the SAMS-S consistently and comparably measures academic motivation among male and female medical students. It is noteworthy that even at the stage of strict invariance, the scale demonstrated good fit, reinforcing its robustness and applicability in both groups (Table 5).

## 4. Discussion

Medical education is a complex and dynamic field, influenced by various factors, necessitating highly motivated students to ensure effective learning. Academic motivation, a crucial factor in student development, is linked to other psychological and demographic aspects. Within medicine, such motivation impacts decisions such as specialty choice and can vary based on socioeconomic and geographic contexts. Self-determination theory provides a framework for understanding motivation in terms of intrinsic, extrinsic, and amotivation. Several tools assess academic motivation, notably the Academic Motivation Scale (AMS). However, to reduce response burden and adapt to different contexts, the Short Academic Motivation Scale (SAMS) was developed, originally for students in the United Kingdom. Considering the unique characteristics of medical students in Peru, it is essential to adapt and validate instruments like the SAMS. This study aimed to evaluate the psychometric properties of the SAMS in Peruvian medical students.

The adaptation and confirmatory factor analysis (CFA) of the Short Scale of Academic Motivation (SAMS-S), based on a seven-dimension structure [35], reveal that all factor loadings, exceeding the 0.50 criterion, confirm the relevance of items to their respective factors. This supports the conceptual structure of the instrument and its applicability in the Peruvian educational context [49]. When analyzing the goodness-of-fit indices, the Peruvian version of the SAMS-S shows promising results, demonstrating a fit comparable, and even superior, to the original English version. Specifically, the CFI (0.95) and TLI (0.92) values for the Peruvian version not only surpass those of the English version (CFI of 0.94 and TLI of 0.90) but also suggest a slightly better fit, potentially reflecting an effective cultural adaptation and the relevance of academic motivation constructs in the Peruvian context [58]. However, the analysis also highlighted challenges related to discriminant validity, especially in the high correlations among some dimensions, suggesting the possible existence of a higher-order factor grouping these related dimensions (Reise et al., 2013; Schwarz et al., 2014) [56,57]. Therefore, evaluating a second model that considers a higher-order factor provides an additional perspective on the scale’s structure, suggesting that the SAMS-S can be interpreted as a multi-level scale. The fit indices of this alternative model, though slightly lower than those of the first-order model, still indicate an adequate fit, offering a more holistic and theoretically coherent view of academic motivation.

In reviewing the reliability of the Peruvian version of the Short Scale of Academic Motivation (SAMS-S), notable differences are identified compared to the original English version [35]. For example, the IMK subscale showed a Cronbach’s alpha (α) of 0.84 in the English version, whereas in the Peruvian version, it recorded a 0.76. This discrepancy, albeit modest, might reflect cultural variations in item interpretation or in the very structure of motivation for knowledge. Despite these differences, both versions of the scale demonstrate acceptable reliability in most of their subscales. Notably, the IDR and IJR subscales in the Peruvian adaptation did not reach the commonly accepted threshold of 0.7 for good reliability [59], contrasting with the English version. However, it is important to consider that while an alpha of 0.70 or higher is preferable for studies underpinning critical decisions, exploratory research may accept a lower value, close to 0.60 [60]. This flexibility suggests that the observed variations might be due to peculiarities of the Peruvian educational context or the adaptation methodology [61]. Focusing on the results of the second-order model of the SAMS-S in the Peruvian context, a general robustness in reliability is observed, not only at the subscale level but also in the total internal consistency of the model. This second-order approach, evaluating reliability through α, ω, and Composite Reliability (CR), not only confirms the solidity of individual dimensions but also reveals an excellent overall reliability with an α of 0.82, an ω of 0.91, and a CR of 0.86. These results emphasize the coherence and integrity of the second-order model, offering a comprehensive view of the structure of academic motivation as a unified construct.

### 4.1. Implications

Understanding academic motivation, particularly in specialized contexts such as medical training, transcends the educational sphere and has significant implications across multiple dimensions. Firstly, intrinsic motivation has been shown to be a crucial factor for academic and professional success. By grasping the drivers of medical students’ motivation, institutions can craft learning experiences that support and enhance this motivation. In the long term, this can lead to more committed and efficient healthcare professionals, thereby improving the overall quality of medical care. Additionally, with the early identification of areas where students show less motivation, professional development programs can be tailored to address these gaps. This is critical in medicine, where continuous learning and adaptation are imperative. Furthermore, medical school admission policies could benefit from considering not only academic excellence but also the levels and types of motivation of candidates. Institutions might, for example, use tools like the SAMS-S to identify applicants who, in addition to having the necessary academic skills, are intrinsically motivated to pursue a career in medicine. Lastly, the gender invariance demonstrated by the SAMS-S underscores the importance of policies ensuring gender equality in medical education. Institutions must strive to ensure that opportunities and resources are equally accessible to everyone, regardless of gender.

### 4.2. Limitations

Despite the promising results and the significant contributions of this research to the existing literature, it is vital to acknowledge and address several limitations that could have influenced the study’s findings. Firstly, due to the cross-sectional nature of the design, relationships between variables can only be established at a specific point in time. This prevents inferring causality or tracing the temporal trajectories of changes in academic motivation. To overcome this limitation, future research could use longitudinal designs, allowing for a more detailed analysis of the trends and shifts in motivation over time. Secondly, the study sample was limited to medical students in Peru, which might restrict the generalization of the findings to other contexts or academic disciplines. It would be beneficial to replicate this study in different geographical contexts and with students from different disciplines to determine if the findings are consistent across a variety of educational settings. Lastly, although a detailed analysis of gender invariance was conducted, other potential sources of variability, such as academic year, region of origin, or socioeconomic level, were not explored. These variables could have a significant impact on academic motivation and should be considered in future research.

## 5. Conclusions

The second-order model of the Short Scale of Academic Motivation (SAMS-S) has proven to be a reliable and valid instrument for measuring academic motivation among medical students in the Peruvian context. The gender invariance demonstrated by the SAMS-S in this study is a particularly notable finding, highlighting the consistency and comparability of the measurement of academic motivation between male and female medical students. The SAMS-S in this specific setting demonstrates not only the applicability of the instrument across different cultural and geographical contexts but also its robustness and reliability in measuring different dimensions of academic motivation. In terms of future research directions, it would be beneficial to expand the application of the SAMS-S to a broader variety of educational and cultural contexts, to further assess its validity and applicability.

## Figures and Tables

**Figure 1 behavsci-14-00316-f001:**
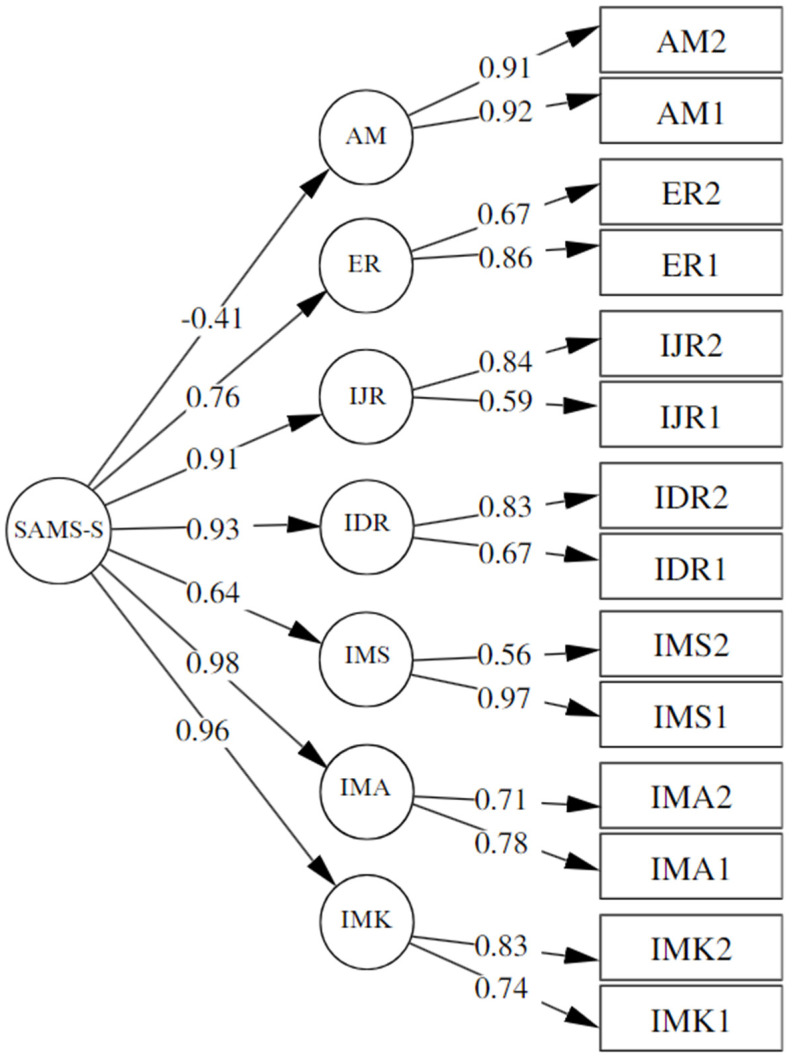
SAMS-S = Academic Motivation; IMK = Intrinsic Motivation to Know; IMA = Intrinsic Motivation toward Accomplishment; IMS = Intrinsic Motivation to Experience Stimulation; IDR = Identified Regulation; IJR = Introjected Regulation; ER = External Regulation; AM = Amotivation.

**Table 1 behavsci-14-00316-t001:** Sociodemographic characteristics.

Characteristics		n	%
Sex	Female	159	59.3
	Male	109	40.7
Region of Origin	Coast	144	53.7
	Jungle	39	14.6
	Highlands	85	31.7
Year of Study	1	142	53.0
	2	48	17.9
	3	35	13.1
	4	17	6.3
	5	24	9.0
	6	2	0.7

**Table 2 behavsci-14-00316-t002:** Descriptive analysis of items.

Subscales	Item	M	SD	g_1_	g_2_
	Estoy cursando esta Carrera…				
IMK1	por el placer que experimento al descubrir cosas nuevas nunca antes vistas.	5.3	1.32	−0.38	−0.67
IMK2	debido a que mis estudios me permiten seguir aprendiendo acerca de muchas cosas que me interesan.	5.48	1.23	−0.32	−0.98
IMA1	por el gusto que siento al superarme en uno de mis logros personales.	5.54	1.26	−0.42	−0.85
IMA2	porque la universidad me permite experimentar una satisfacción personal en mi búsqueda de la excelencia en mis estudios.	5.3	1.16	−0.13	−0.93
IMS1	por el placer que experimento al leer autores interesantes.	4.79	1.36	0	−0.75
IMS2	por el placer que siento al estar completamente absorto por lo que ciertos autores han escrito.	4.45	1.35	−0.05	−0.24
IDR1	porque creo que una educación universitaria me ayudará a prepararme mejor para la carrera que he elegido.	5.41	1.35	−0.38	−0.87
IDR2	porque eventualmente me permitirá ingresar al mercado laboral en un campo que me gusta.	5.36	1.29	−0.35	−0.66
IJR1	debido al hecho de que cuando tengo éxito en la universidad, me siento importante.	4.96	1.31	−0.03	−0.81
IJR2	porque quiero demostrarme a mí mismo que puedo tener éxito en mis estudios.	5.45	1.32	−0.43	−0.82
ER1	para obtener un trabajo más prestigioso en el futuro.	5.25	1.36	−0.38	−0.83
ER2	con el fin de tener un mejor salario en el futuro.	4.87	1.43	−0.1	−0.97
AM1	porque no entiendo por qué voy a la universidad y, francamente, no me importa.	2.13	1.69	1.17	−0.16
AM2	porque no lo sé; no puedo comprender qué hago en la Universidad.	2.08	1.62	1.25	0.15

Note: IMK = Intrinsic Motivation to Know, IMA = Intrinsic Motivation toward Accomplishment, IMS = Intrinsic Motivation to Experience Stimulation, IDR = Identified Regulation, IJR = Introjected Regulation, ER = External Regulation, AM = Amotivation, M = Mean, SD = Standard Deviation, g_1_ = skewness, g_2_ = kurtosis.

**Table 3 behavsci-14-00316-t003:** Confirmatory factor analysis and reliability of the first model.

Items	IMK	IMA	IMS	IDR	IJR	ER	AM
IMK1	0.72						
IMK2	0.85						
IMA1		0.77					
IMA2		0.71					
IMS1			0.91				
IMS2			0.60				
IDR1				0.70			
IDR2				0.80			
IJR1					0.60		
IJR2					0.83		
ER1						0.89	
ER2						0.65	
AM1							0.85
AM2							0.99
SAMS-S Correlation							
AVE	0.62	0.55	0.59	0.57	0.52	0.61	0.85
IMK	-	0.03	0.14	0.12	0.18	0.22	0.01
IMA	−0.17	-	0.74	0.52	0.55	0.48	0.23
IMS	−0.38	0.86	-	0.62	0.98	0.76	0.45
IDR	−0.34	0.72	0.79	-	0.98	0.79	0.35
IJR	−0.42	0.74	0.99	0.99	-	1.00	0.52
ER	−0.47	0.69	0.87	0.89	1.00	-	0.44
AM	0.10	0.48	0.67	0.59	0.72	0.66	-
Internal Consistency							
α	0.76	0.71	0.71	0.72	0.67	0.73	0.91
ω	0.76	0.71	0.74	0.72	0.69	0.75	0.92
CR	0.76	0.71	0.74	0.72	0.69	0.75	0.92

Note: IMK = Intrinsic Motivation to Know, IMA = Intrinsic Motivation toward Accomplishment, IMS = Intrinsic Motivation to Experience Stimulation, IDR = Identified Regulation, IJR = Introjected Regulation, ER = External Regulation, AM = Amotivation, α = Cronbach’s Alpha, ω = McDonald’s Omega, CR = Composite Reliability, AVE = Average Variance Extracted, below the diagonal = interfactorial correlations; above the diagonal = square of the correlation (φ^2^).

**Table 4 behavsci-14-00316-t004:** Reliability of the second model (second-order model) total internal consistency.

Dimensions	IMK	IMA	IMS	IDR	IJR	ER	AM	General
α	0.76	0.71	0.71	0.72	0.67	0.73	0.91	0.82
ω	0.76	0.71	0.76	0.72	0.69	0.74	0.91	0.91
CR	0.76	0.71	0.76	0.72	0.69	0.74	0.91	0.86

Note: IMK = Intrinsic Motivation to Know; IMA = Intrinsic Motivation toward Accomplishment; IMS = Intrinsic Motivation to Experience Stimulation; IDR = Identified Regulation; IJR = Introjected Regulation; ER = External Regulation; AM = Amotivation.

**Table 5 behavsci-14-00316-t005:** Gender invariance.

Invariance	χ^2^	df	*p*	TLI	RMSEA	SRMR	CFI	ΔCFI
Configural	301	140	<0.001	0.872	0.093	0.069	0.901	
Metric	317	147	<0.001	0.871	0.093	0.079	0.896	0.005
Scalar	325	154	<0.001	0.876	0.091	0.08	0.895	0.001
Strict	351	168	<0.001	0.878	0.09	0.079	0.887	0.008

Note: M1 = Configural; M2 = Metric; M3 = Scalar; M4 = Strict; χ^2^: Chi-Square; df = Degrees of Freedom; RMSEA = Root Mean Square Error of Approximation; SRMR = Standardized Root Mean-Square; TLI = Tucker-Lewis Index; CFI = Comparative Fit Index; ΔCFI = Comparative Fit Index difference.

## Data Availability

Data can be requested from the corresponding author.

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
