# Peer review of "Psychometric Properties of a Short Academic Motivation Scale (SAMS) in Medical Students"

_behavsci, 2024, doi:10.3390/bs14040316_

Round 1
Reviewer 1 Report
Comments and Suggestions for Authors
The main goal of this study was to validate an SDT measure with Peruvian medical students. It is strictly a validation study with a new sample, so the focus of the intro, methods, and discussion should be all about validity. There is currently too much talk in the introduction and discussion sections about how important motivation is for medical students. That is not the focus of your study, and should be dropped from the manuscript.
There are three big questions that the manuscript does not currently address, but should. First, is there something different about Peruvian medical students that would lead you to suspect the results would differ for them compared to Canadian students? No information is provided about the Peruvian context, and no hypotheses are offered about potential differences from other samples. Second, does it make sense to ask about the motivation of something as broad as attending medical school? The stem for these items is "I'm studying for this career because..." Looking at the means of the items, it appears that students rate just about all forms of motivation, from intrinsic to external, about the same. This may be evidence that these items are too general to really differentiate between the different forms of regulation that SDT covers. Third, although the CFA produced 7 distinct scales, do we really think these are distinct phenomenologically? According to the correlations in Table 3, the scales are very strongly correlated with each other. IMK is correlated with IJR at .94. One of the reported correlation coefficients is over 1.0! I'm not sure how that happened. It looks like all of these subscales would fit into a single higher-order factor that would have strong reliability.
I believe the paper would need to be reframed as a straight measurement validation piece and these three issues would need to be addressed before this paper could be accepted for publication.
Author Response
The main goal of this study was to validate an SDT measure with Peruvian medical students. It is strictly a validation study with a new sample, so the focus of the intro, methods, and discussion should be all about validity. There is currently too much talk in the introduction and discussion sections about how important motivation is for medical students. That is not the focus of your study, and should be dropped from the manuscript.
Response: Thank you for your recommendations. Corrections have been made to the introduction and discussion according to what was indicated.
There are three big questions that the manuscript does not currently address, but should. First, is there something different about Peruvian medical students that would lead you to suspect the results would differ for them compared to Canadian students? No information is provided about the Peruvian context, and no hypotheses are offered about potential differences from other samples.
Response: Thank you for your comments. A paragraph has been added highlighting some fundamental similarities in academic motivation among medical students globally; however, Peruvian students may exhibit unique particularities due to contextual and cultural factors. For example, the transition to virtual education during the COVID-19 pandemic underscores the resilience and adaptability of Peruvian medical students, a challenge that, although global, may have had distinct nuances in Peru due to differences in technological infrastructure, access to educational resources, and institutional support compared to Canada. Additionally, intrinsic motivation linked to values such as public service may be particularly strong among Peruvian students, influenced by the socioeconomic context and specific health needs of the region. This could result in differences in how motivational factors, such as self-efficacy and professional aspirations, influence their academic performance and specialization decisions. “In Peruvian students, academic motivation reveals complexities inherent in the educational process in various areas of study. In this sense, both intrinsic motivation, linked to the personal desire to learn, and extrinsic motivation, associated with external rewards such as job opportunities, influence academic performance, albeit with weak correlations (Yarin et al., 2022). In contexts such as that of international business students, social pressure and personal aspirations play a significant role in career choice, differentiating from fields like health, where motivation for public service predominates (Vicente-Ramos et al., 2020). In the field of medicine, the trend indicates high motivation, particularly notable among women, despite the challenges presented by the transition to virtual education during the COVID-19 pandemic. This underscores the need for further research to thoroughly understand the factors that affect motivation in medical education scenarios (Rivadeneyra-Zeña & Ñique-Carbajal, 2023). Additionally, adaptation to university life emerges as a significant challenge, where elements such as procrastination, self-esteem, and self-efficacy are identified as critical determinants for academic success, highlighting the importance of addressing these factors to facilitate better adaptation and student performance (Hernández et al., 2020). ”
Second, does it make sense to ask about the motivation of something as broad as attending medical school? The stem for these items is "I'm studying for this career because..." Looking at the means of the items, it appears that students rate just about all forms of motivation, from intrinsic to external, about the same. This may be evidence that these items are too general to really differentiate between the different forms of regulation that SDT covers.
Response: It is important to highlight that motivation is a complex and multifaceted construct, encompassing both intrinsic and extrinsic aspects. The use of an already established scale in English and its subsequent validation in the Peruvian context responds to the need for culturally adapted and methodologically solid measurement tools to investigate this phenomenon in a specific environment. The presence of homogeneous responses across various forms of motivation might reflect the integration of multiple sources of motivation in the student experience, but it also highlights the need to explore the dimensions of motivation more deeply to obtain clearer and more significant differentiations. The validity of the scale in the Peruvian context, therefore, is not only supported by the translation and adaptation of the instrument but also by its ability to reflect and effectively measure the complexities and particularities of motivation among medical students in Peru. Cultural validation of the measurement instrument ensures its relevance and accuracy in specific contexts.
Third, although the CFA produced 7 distinct scales, do we really think these are distinct phenomenologically? According to the correlations in Table 3, the scales are very strongly correlated with each other. IMK is correlated with IJR at .94. One of the reported correlation coefficients is over 1.0! I'm not sure how that happened. It looks like all of these subscales would fit into a single higher-order factor that would have strong reliability.
Response: Thank you for your recommendation, corrections were made, and the second-order model was established.

Reviewer 2 Report
Comments and Suggestions for Authors
The main finding reported in this paper indicates that the Spanish version of the Short Academic Motivation Scale for Peruvian medical students has adequate psychometric properties.
The title clearly and precisely reflects the study and findings, and the abstract is written in a clear and comprehensive way.
The introduction presents the study in an appropriate context, and the background of the study, including previous literature, is adequately presented. However, I would suggest a more balanced, comprehensive, and critical view of earlier studies in the area of academic motivation.
The methods and data are clearly described, and they are used adequately to address the research questions posed, while the statistical methods used seem valid. The study has been conducted in conformity with the ethical standards of the field, and the author(s) identify the committee approving the study. The author(s) checked data distribution (i.e., Skewness and Kurtosis), but they should provide other pre-analyses, such as linearity and outliers, missing and influential cases and how they were treated, as well as the critical values of multivariate normality (e.g., Mardia’s coefficient or multivariate Kurtosis). Of course, authors should justify the use of the tests and explain whether their data conform to the assumptions of the tests. In addition to the internal consistency, the composite reliability of the construct and the average variance extracted should also be calculated.
The results are presented appropriately. Nevertheless, they should be improved based on the required and recommended changes to methods and data.
The discussion and conclusion address the research objective, and the results are interpreted in light of previous knowledge. The conclusion is justified.
Author Response
The introduction presents the study in an appropriate context, and the background of the study, including previous literature, is adequately presented. However, I would suggest a more balanced, comprehensive, and critical view of earlier studies in the area of academic motivation.
Response: Thank you for your recommendations. Corrections were made throughout the introduction.
The methods and data are clearly described, and they are used adequately to address the research questions posed, while the statistical methods used seem valid. The study has been conducted in conformity with the ethical standards of the field, and the author(s) identify the committee approving the study. The author(s) checked data distribution (i.e., Skewness and Kurtosis), but they should provide other pre-analyses, such as linearity and outliers, missing and influential cases and how they were treated, as well as the critical values of multivariate normality (e.g., Mardia’s coefficient or multivariate Kurtosis). Of course, authors should justify the use of the tests and explain whether their data conform to the assumptions of the tests. In addition to the internal consistency, the composite reliability of the construct and the average variance extracted should also be calculated.
Response: We made the suggested changes in the methods and results sections.
The results are presented appropriately. Nevertheless, they should be improved based on the required and recommended changes to methods and data. The discussion and conclusion address the research objective, and the results are interpreted in light of previous knowledge. The conclusion is justified.
Response: Thank you for the feedback; corrections were made according to the re-evaluation.
A document marked in red is attached for verification of the corrections.
